# Current Approaches to the Treatment of Gastric Varices: Glue, Coil Application, TIPS, and BRTO

**DOI:** 10.3390/medicina55070335

**Published:** 2019-07-03

**Authors:** Vedat Goral, Nevin Yılmaz

**Affiliations:** 1Department of Gastroenterology, School of Medicine, Istanbul Medipol University, Istanbul–34214, Turkey; 2Department of Gastroenterology, School of Medicine, Near East University, Nicosia-99138, Cyprus

**Keywords:** gastric varices, bleeding, glue and coil application, BRTO

## Abstract

Gastric varices are less common than esophageal varices, and their treatment is quite challenging. Gastric varix bleedings (GVB) occur less frequently than esophageal varix (EV) bleedings and represent 10to 30% of all variceal bleedings. They are; however, more severe and are associated with high mortality. Re-bleeding may occur in 35to 90% of cases after spontaneous hemostasis. GV bleedings represent a serious clinical problem compared with esophageal varices due to their location. Sclerotherapy and band ligation, in particular, are less effective. Based on the anatomic site and location, treatment differs from EV and is categorized into two groups (i.e., endoscopic or radiologic treatment). Surgical management is used less frequently. Balloon-occluded retrograde transvenous obliteration (BRTO) and cyanoacrylate are safe but there is a high risk of re-bleeding. Portal pressure elevates following BRTO and leads to worsening of esophageal varix pressure. Other significant complications may include hemoglobinuria, abdominal pain, fever, and pleural effusion. Shock and atrial fibrillation are major complications. New and efficient treatment modalities will be possible in the future.

## 1. Introduction

Gastroesophageal varices (GOV) occur in approximately 50% of patients with liver cirrhosis. Gastric varices are dilated collateral blood vessels that develop as a complication of portal hypertension. The best known and main cause of portal hypertension is liver cirrhosis and there is an increase in resistance through hepatic sinusoids. Gastric varices are less common than esophageal varices (EV), and bleeding from gastric varices represent 10 to 30% of GOV bleedings [1,2,3,4]. However, 35 to 90% of patients with gastric variceal bleeding (GVB) experience re-bleeding, and GVBs tend to be more severe with higher mortality. Mortality and morbidity rates vary between 30 and 50%, depending on the severity of the underlying condition, varix size, and presence of signs that predict bleeding. The more aggressive interventions, such as evaluation of liver transplantation or a transjugular intrahepatic portosystemic shunt (TIPS) can cause high mortality and morbidity risk. [5].

Gastric varices, according to Sarin, were classified as gastroesophageal or isolated gastric varices (IGVs), based on the relationship with esophageal varices and the location in the stomach. Type 1 GOV is localized in the lesser curvature, whereas type 2 GOV is located in the fundus and continues along the greater curvature. Type 1 IGVs exist in the gastric fundus and they do not extend into the esophagus or the cardia. Type 2 IGV is defined as ectopic variations that occur in other parts of the stomach. [2]. (see Figure 1).

Gastroesophageal varices and esophageal varices share the same anatomy, pathophysiology, and blood supply. They usually receive blood from the left gastric vein and lamina propria. Isolated gastric varices, on the other hand, stem from underneath the lamina propria in the submucosa and are found usually in the fundus. Their blood comes from short and posterior gastric veins. IGVs frequently present with large gastrorenal shunts [6]. Therefore the localization of gastric varices is an important factor when deciding on a treatment.

## 2. Treatment

There are some therapeutic guidelines, such as AASLD (American Association for the Study of Liver Disease), the Baveno Consensus Workshop, and NICE (the National Institute for Health and Care Excellence). Based on the anatomic site and location, treatment differs from EV and is categorized into two groups (i.e., endoscopic or radiologic treatment). Surgical management is used less frequently.

(1)In endoscopic treatment gastric variceal sclerotherapy, gastric variceal band ligation (GVL), glue application with varix occlusion, endoscopic ultrasound (EUS)-guided coil/glue application, thrombin applications or a combination is used.(2)Radiologic treatments include TIPS (transjugular intrahepatic portosystemic shunt) and BRTO (balloon retrograde transvenous obliteration).

*Endoscopic treatments* include sclerotherapy, band ligation, cyanoacrylate injection, or their combinations. In acute gastric variceal bleeding, endoscopy can be planned following initial resuscitation (stabilization, antibiotic therapy, terlipressin administration, etc.). Endoscopy is performed to identify whether the bleeding site is the fundus or the gastroesophageal junction. Generally, glue treatment is recommended for fundal variceal bleedings, while glue or band ligation is recommended for bleedings at the gastroesophageal junction. (Figure 1). Once GVB is controlled, variceal scans are performed at certain intervals [6]. Band ligation is now preferred over sclerotherapy in GOV type 1, and is not recommended in IGV because these varices occur in the fundus, greater curvature and submucosa. If the varix and contralateral wall cannot be captured in band ligation, blood flow remains uninterrupted and may lead to massive bleeding. The standard treatment for IGV worldwide is to administer a sclerotizing substance in the form of glue injection. Once the glue is injected, it polymerizes within the varix and forms a hard, firm structure. If variceal bleeding cannot be controlled, then balloon tamponade, a TIPS, surgical shunt, or liver transplantation is scheduled. However, there is a 15 to 20% risk of re-bleeding during the first five days, even if optimal treatment was given.

*Gastric variceal sclerotherapy* yields greater success in esophageal varices but is less efficient in gastric varices. More sclerotizing substance is required as the varix is larger Also more sclerosant is used for sclerotherapy particularly in GOV2 and IGV1 compared with GOV1. Fever, retrosternal and abdominal pain, and widespread ulcerations may occur as complications due to the procedure. Serious complications including perforation and mediastinitis may also develop, and may lead to mortality in about 50% of the cases [6].

*Gastric variceal band ligation* is not the initial medical treatment of choice in gastric variceal bleedings. Risk of re-bleeding is quite high due to supplying vessels. Repeat endoscopy and endoscopic variceal ligation (EVL) should be performed in 1–2 week intervals, and monitoring should be maintained unless the varix shrinks completely. EVL and injection sclerotherapy (EVLIS) may also be used in combination. One ml of ethanolamine oleate 5% is administered intravariceally above the location intended for band ligation [7,8]. Success rate with EVLIS is 89%, but there is a 33% risk of re-bleeding. Therefore, due to the required experience and processing time as well as the risk of iatrogenic trauma, it is not recommended.

*Thrombin treatment* functions by catalyzing the conversion of fibrinogen to fibrin and supporting coagulation [9,10,11,12]. It also has an effect on platelets. This is a treatment modality that has been known for a long time but is used infrequently. It is an alternative for cyanoacrylate given the fewer complications and side effects. Generally, 1 mL of thrombin is injected into varices. Five ml of thrombin solution (Floseal, Baxter) is potent enough to coagulate 1 liter of blood in less than 60 seconds. Thrombin treatment is very beneficial in difficult-to-treat GOV2 varices.

*Cyanoacrylate (glue) treatment* uses N-butyl-2-cyanoacrylate. Following administration, cyanoacrylate is polymerized and exerts its effects instantly within 20 s [13,14,15,16,17,18,19,20,21,22,23,24]. The fatty contrast agent lipiodol ultra-fluid (Therapex, Canada) is used to avoid occlusion in the endoscopy channel during the procedure. It is recommended to use a 50:50 mixture of cyanoacrylate and lipiodol (or normal saline). Treatment with this approach also reduces the risk of embolization. Because normal gastroscopies have a 3.2 mm working width, this procedure does not allow for the passage of injection catheter in retroflexion and is not suitable for this method; therefore, endoscopy instruments with a 3.7 mm working channel should be used. Glue and lipiodol are diluted 1:1 to 1:1.6, with an increased risk of embolism at higher dilutions. Medicinal products called Dermabond and Glubran do not require lipiodol due to slow polymerization. The medicinal products used and their properties are provided in Table 1.

There are some risks and potential complications associated with cyanoacrylate injection. For example, pulmonary embolism (in pulmonary veins), acute renal damage (in renal veins), splenic, or portal vein thrombosis (may subsequently necessitate liver transplantation) may occur. Cyanoacrylate may less frequently lead to gastric ulcer (0.1%), major gastric varix bleeding, mesenteric hematoma, hemoperitoneum, bacterial peritonitis, etc. [12,13]. Antibiotic treatment may; therefore, be initiated where this is deemed necessary.

*EUS-guided glue and coil treatments* are administered successfully and safely. Varices are observed clearly by performing color doppler EUS prior to treatment [25]. The lumen is filled with water for better visualization of gastric fundal varices. EUS should be used in anterograde fashion. Diaphragmatic crus muscle layer should be seen between the esophageal wall and gastric fundal varix. A 19 gauge, 0.035 inch needle is inserted into the varix transesophageally, followed by insertion of coils. The most popular coils are Tornado- and Nester-type coils (Figure 2a,b). These coils have a diameter of 8–20 mm and have fibrillary structures on them, to which platelets adhere, where they coagulate and cause closure within the varix. The diameter of the coil to be used is calculated with EUS. A 20% excess is calculated over the varix diameter. The stylet is pulled back and the coils are pushed into the varix with the assistance of stylet. Immediately after coil application, 1 ml of 2-Octyl-cyanoacrylate is administered into the varix with the same needle. Varix obliteration rate with glue is 94.7% compared to 90.9% with EUS-guided cyanoacrylate application [12,13,14]. Controls are performed one month later, usually demonstrating disappearance of varices. Some complications may occur in 57.9% of the patients treated with the glue method and in 9.1% of those for whom endoscopic cyanoacrylate was used. Interestingly, CT shows embolism in 9.1% of patients treated with the glue method but no symptoms are observed in 80% of these patients [13,14].

*Hemospray application* is a procedure performed using the hemospray device. Hemospray, unlike traditional therapies, is a nonthermal, nontraumatic, noncontact therapeutic modality that does not require the precise targeting of other endoscopic devices. It is used with success, especially in patients who received glue or cyanoacrylate treatment and suffered from bleeding [4,5].

*Radiologic (endovascular) treatments* include a transjugular intrahepatic portosystemic shunt (TIPS) and BRTO or a TIPS + BRTO combination [18,19,20,21,22].

*A transjugular intrahepatic portosystemic shunt*, or TIPS, is a procedure that creates a shunt between portal vein and hepatic venous systems via transjugulary venous way placing a stent in the liver parenchyma at the interventional radiology department [18,21]. A TIPS reduces esophageal variceal bleeding and treat complication with liver cirrhosis, such as refractory ascites. For a TIPS, access is gained via the transjugular veins, when a metal-coated expandable stent is inserted in the liver with v. hepatica end branch and v. porta end branch (Figure 3). Thus, portal pressure is reduced, leading to reduced varix pressure and, therefore, reduced bleeding. This procedure should be performed by experienced professionals in interventional radiology centers.

Moderate sedation may be used. Dye (contrast material) is then injected into the vein and the balloon is inflated to place the stent. The coated stent is then used to connect the portal vein to one of the hepatic veins. At the end of the procedure, portal vein pressure goes down. The catheter with the balloon is then removed. The total procedure takes about 60 to 90 min to complete. According to the studies, the one-year patency rate of the coated stent is 85%. A TIPS also improve quality of life in cirrhosis patients [18,21]. There are some complications of this procedure, such as the possibility of hepatic encephalopathy and bleeding.

Balloon-occluded retrograde transvenous obliteration (BRTO) application is a popular modality in Asian countries. Gastric varices are associated with gastrorenal shunt (GRS) in 60 to 85% of cases. Flow in GRS is into the systemic circulation. BRTO is an endovascular technique used as an adjunctive or therapeutic alternative to a TIPS for the treatment of gastric varices [17,18,19,20,21]. Radiologists first block GRS using the balloon method in interventional radiology units. This is followed by intravenous administration of sclerotizing substances. (Figure 4). [26]. BRTO achieves success in 76.9 to 100% of cases in acute gastric varices. Likelihood re-bleeding is low with 0to 14%. BRTO is as successful as a TIPS without increasing hepatic encephalopathy potential.

We recommend reviewing computed tomography or magnetic resonance images prior to the procedure to determine the approach that provides the best angle and to decide on a shunt. The sclerotizing substances used for BRTO are ethanolamine oleate, polidocanol sodium tetradecyl sulfate foam, and polidocanol foam. Ethanolamine oleate was described as the primary sclerotizing substance used in this procedure. Recently, foam scleroses have gained popularity as alternative embolization agents that offer better varix wall contact and sclerotizing activity potential at a lower dose. BRTO is used in interventional radiology, and the right femoral vein approach was preferred in most reported cases. However, some centers have adopted only the jugular vein approach. Catheterization of the gastrorenal shunt via the left renal vein can be achieved using a catheter with an occlusion balloon. Reverse balloon catheters offer easier and more consistent access to the shunt. The approach we recommend involves the insertion of an appropriate size access sheath within the inferior vena cava (IVC) or distal renal vein (6–12 Fr) and a 5 Fr diagnostic catheter is then used to choose the left renal vein. The diagnostic catheter is then changed to an angled-end catheter, which is used to choose the shunt. Alternatively, a 5 Fr Simmons II catheter can be used to choose the left renal vein. It is pulled back until they are aligned edge-to-edge. Then a 0.035 inch hard wire is advanced into the shunt as far as it goes, after which an occlusion balloon catheter with a diameter of 8.5 to 32 mm is placed. Injection of the sclerotizing substance is performed with or without a microcatheter for a more selective injection. The anatomy of the venous drainage pattern is very important. If there are multiple small draining veins without definable shunts, BRTO cannot be used. [27].

For portal venous shunts, BRTO and cyanoacrylate are safe but there is a high risk of re-bleeding. Portal pressure elevates following BRTO and leads to worsening of esophageal varix pressure. Other significant complications may include hemoglobinuria, abdominal pain, fever, and pleural effusion. Shock and atrial fibrillation are major complications. [28].

The efficacy of secondary prophylaxis of gastric variceal bleed has not been well studied in the literature. For this reason, non-selective β-blocker drugs is used for secondary prophylaxis. A comparison of the efficacy of β-blocker treatment and cyanoacrylate injection for the prevention of gastric variceal re-bleeding was carried out in some articles [1]. A randomized controlled trial of 158 patients was conducted to examine the role of simvastatin in combination with standard therapy for reducing the risk of further variceal hemorrhage [23]. However, it has not been proven to be very effective for secondary prophylaxis.

In summary, gastric variceal bleeding is associated with higher mortality and more severe progression, and is less common than EV bleeding. Bleeding often recurs after spontaneous hemostasis. The treatment of gastric varices is quite difficult and sclerotherapy and band ligation application, specifically, are less effective. The type of gastric varicose and the type of shunt are important factors when deciding on an interventional treatment. In regard to the overall cost of gastric variceal therapy, coils + EUS-guided cyanoacrylate is a more cost-effective method than the use of cyanoacrylate injection alone. Furthermore, new endoscopic (e.g., the Gore Viatorr flexible, self-expanding, implantable TIPS endoprosthesis-reduced permeability with expanded polytetrafluoroethylene (ePTFE) graft lining) and radiologic (EUS–guided angiotherapy) combination treatments may be a way to improve outcomes or decrease complications associated with acute GV bleeding [24]. The recent interventional methods need further study but new treatment approaches will be available in the near future.

## Figures and Tables

**Figure 1 medicina-55-00335-f001:**
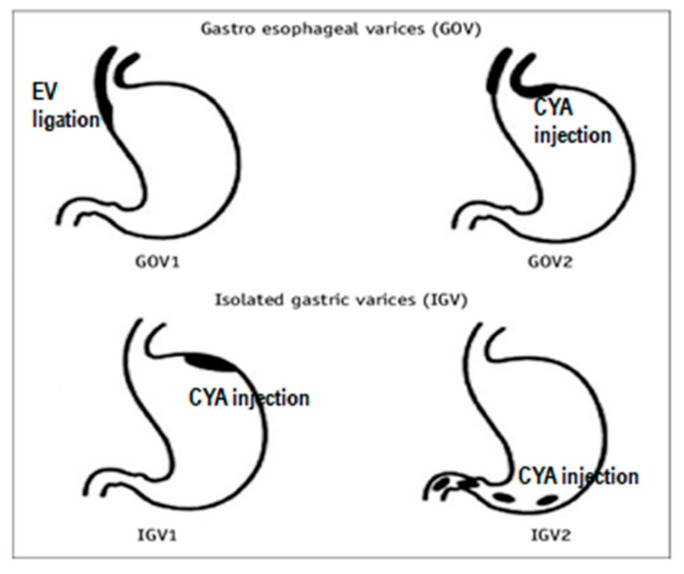
Treatments recommended for gastric varices according to Sarin classification; EVL—endoscopic variceal ligation, CYA—α-cyanoacryrate glue, GOV—gastroesophageal varices, IGV—isolated gastric varices.

**Figure 2 medicina-55-00335-f002:**
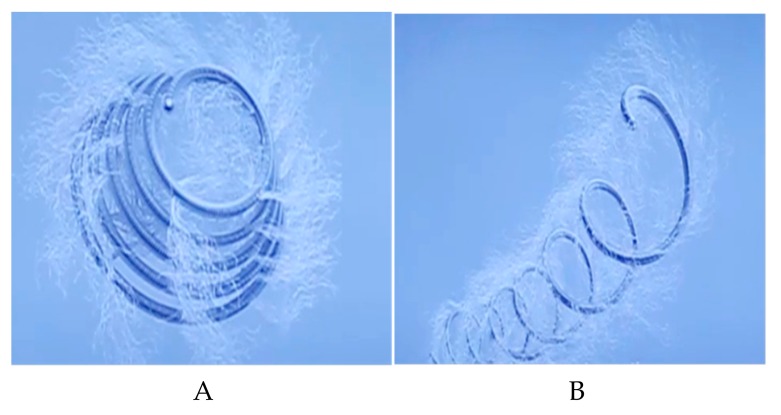
The type of coils used for EUS guıded coil application for varix obliteration. (**A**) Tornado-type coil. (**B**) Nester-type coil.

**Figure 3 medicina-55-00335-f003:**
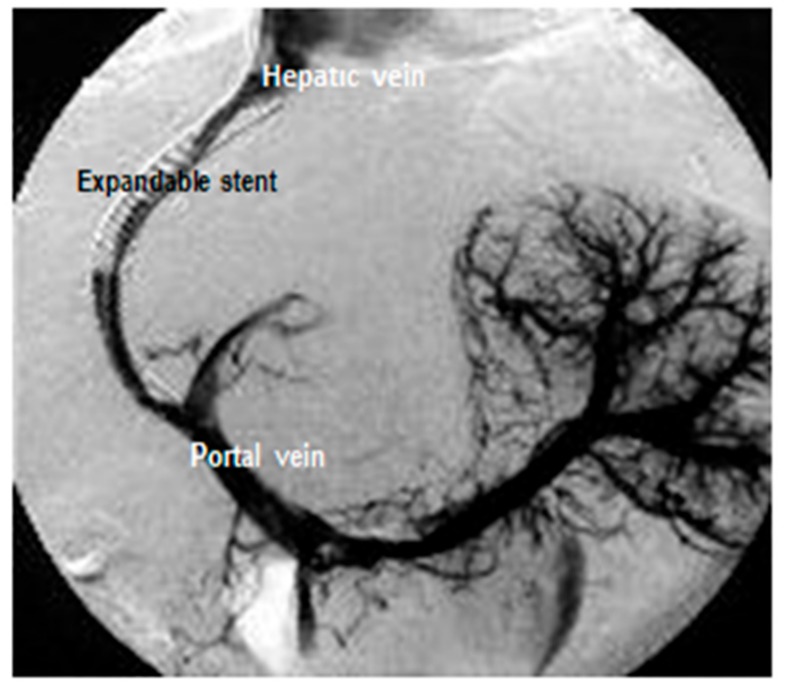
TIPS application.

**Figure 4 medicina-55-00335-f004:**
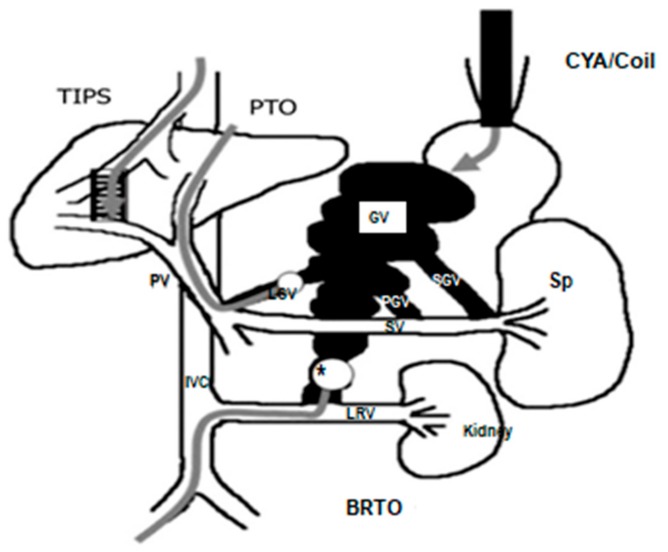
Basic porto-systemic venous anatomy of GV with the shunts and treatment modalities; CYA—α-cyanoacryrate glue, TIPS—transjugular intrahepatic portosystemic shunt, PTO—percutaneous transhepatic obliteration, BRTO—balloon-occluded retrograde transvenous obliteration, IVC—inferior venacava, LGV—left gastric vein, SGV—short gastric vein, PGV—posterior gastric vein, LRV—left renal vein, PV—main portal vein, SV—splenic vein, GV—gastric varices; Sp—spleen. *Conventional BRTO procedure through transfemoral approach with a balloon in the gastrorenal shunt. Modified from [26].

**Table 1 medicina-55-00335-t001:** Medicinal products used in glue treatment and their properties.

Trade Name	Manufacturer	Active Component	Drug	Polymerization Rate	Lipiodol Requirement
Indermil	Covidien	N-butyl-2-cyanoacrylate	05. mL/liquid amp	Fast	yes
Cyanoacrylate	TissueSeal	N-butyl-2-cyanoacrylate	05. mL/liquid amp	Fast	yes
Dermabond	Ethicon	2-Octyl- cyanoacrylate	05. mL/liquid amp	Slow	no
Glubran 2	GEM, Italy	N-butyl-2-cyanoacrylate + methacryloyloxy sulfolane	0.25 mL–05 mL and 1 mL/liquid amp	Slow	no

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
