# Peer review of "Current Approaches to the Treatment of Gastric Varices: Glue, Coil Application, TIPS, and BRTO"

_medicina, 2019, doi:10.3390/medicina55070335_

Round 1

Reviewer 1 Report

- Introduction section: please include the etiology of gastroesophageal varices

- Summarize the risk of mortality and morbidity

- Report on the basis of the current international guidelines, the treatment of varices. Not only endoscopic but also therapeutic

- Conclusion section: briefly sumamrize the cost of different options and the impact in clinical practice

- Report a five-years view on the next future in the treatment of varices

Author Response

Dear Reviewer,

Thank you very much for reviewing mine review. We made some corrections and sending again for publication.

Prof. Dr. Vedat Goral.

Reviewer 2 Report

Drs' Goral and Yilmaz have written a narrative review of techniques to control gastric variceal bleeding. 

My major criticism of this paper is that whilst the section on the application of BRTO is very detailed in terms of description of technique there is very little detail on the application of TIPS. TIPS is more widely used for treatment of GOV worldwide and there have been a number of advances in TIPS application in the last years (variable diameter stents, coils at time of TIPS and use in portal vein thrombosis. Indeed TIPs is not even mentioned in the abstract. There is plentiful data on the outcomes of TIPs in GV bleeding which should be included to allow the reader a meaningful comparison to BRTO and Glue.

I found the section on EUS guided treatments quite confusing. The authors state that varix obliteration with glue is 94.7% compared to 90% with EUS-guided cyanoacrylate application - it should be noted that the in reference supporting this statement (Fujii-lau et al) only 4 patients underwent coil injection followed by glue. Surely this is a limitation that should be acknowledged in the text. 

In the text figure 3 is cited as describing the appearance of injected glue within a varix yet the figure is actually of coils. 

The authors highlight some safety concerns with glue  - there are a number of papers that describe modifications to the technique to improve safety and these should be discussed along with a more detailed account of the technique

Minor comments:

There are a number of grammatical errors. Varicella in line 54 page 2 should read variceal

There is no figure 4 

serum saline (line 91 page 4 ) should be normal saline

There is no mention of secondary prophylaxis of gastric varices in the whole manuscript which is a crucial component of management

Author Response

Dear Reviewer,

Thank you very much for reviewing of mine review.

The mistakes were changed by me.

I mentioned broadly at mine review.

There is Figure 4 at mine review.

Varicella word and serum saline has changed as variceal and normal saline by me.

I am sending mine review after revision.

We are looking forward to see mine article at he journal.

Sincerely yours.

Prof. Dr. Vedat Goral.

Istanbul Medipol University. Turkey.

Round 2

Reviewer 2 Report

The manuscript is improved but there are still some minor errors to be addressed.

Firstly, I do not think there is need to duplicate the figure relating to the Sarin classification.

Secondly, in the text figure 3 is said to describe the appearance of glue polymerisation within the varix. The figure actually shows examples of coils for embolisation. 

Author Response

Dear Reviewer,

At first, I want to thank you for correcting my article mistakes.

1) I wrote ' the etiology of gastroesophageal varices at the introduction section.

2) I added current international therapeutic guidelines at 47. sentences.

3) I added  the cost of different options and the impact in clinical practice at 215. sentences.

4) I addede next future in the treatment of varices at 217-220. sentences.

Thank you again.

Sincerely yours.

Prof. Dr. Vedat Goral